# Teaching Urology to Undergraduates: A Prospective Survey of What General Practitioners Need to Know

**DOI:** 10.3390/ijerph182111687

**Published:** 2021-11-07

**Authors:** Ángel Borque-Fernando, Cristina Redondo-Redondo, Concepción Orna-Montesinos, Luis Mariano Esteban, Sophia Denizón-Arranz, Arlanza Tejero-Sánchez, Ramiro García-Ruiz, José Manuel Sanchez-Zalabardo, Jesús Gracia-Romero, Antonio Monreal-Híjar, María Jesús Gil-Sanz, Gerardo Sanz, Mónica Sanz-Pozo, Francisco Romero-Fernández

**Affiliations:** 1Department of Surgery, Gynaecology and Obstetrics, Urology Area, School of Medicine, University of Zaragoza, 50009 Zaragoza, Spain; cristina.redondo.r@gmail.com (C.R.-R.); jgracia1952@gmail.com (J.G.-R.); mjgilsa@salud.aragon.es (M.J.G.-S.); romerouro@gmail.com (F.R.-F.); 2Instituto de Investigación Sanitaria Aragón, 50009 Zaragoza, Spain; arlanza_tejero@hotmail.com (A.T.-S.); ramiro1986@hotmail.com (R.G.-R.); drsanchezurologo@gmail.com (J.M.S.-Z.); mosanzdelpozo@hotmail.com (M.S.-P.); 3Department of English and German Philology, School of Education, University of Zaragoza, 50009 Zaragoza, Spain; conorna@unizar.es; 4Department of Applied Mathematics Engineering, School of la Almunia, University of Zaragoza, 50009 Zaragoza, Spain; 5Faculty of Health Sciences, University Francisco de Vitoria, 28223 Madrid, Spain; sophia.denizon@ufv.es; 6Department of Medicine, Psychiatry and Dermatology, Medicine Area, School of Medicine, University of Zaragoza, 50009 Zaragoza, Spain; amonreal@salud.aragon.es; 7Department of Statistical Methods and Institute for Biocomputation and Physics of Complex Systems-BIFI University of Zaragoza, 50009 Zaragoza, Spain; gerardo@unizar.es

**Keywords:** urological training needs, urological knowledge and skills, practitioners views, undergraduate medical degree, curriculum development

## Abstract

Background: Higher education training in Medicine has considerably evolved in recent years. One of its main goals has been to ensure the training of students as future adequately qualified general practitioners (GPs). Tools need to be developed to evaluate and improve the teaching of Urology at the undergraduate level. Our objective is to identify the knowledge and skills needed in Urology for the real clinical practice of GPs. Methods: An anonymous self-administered survey was carried out among GPs of Primary Care and Emergencies which sought to evaluate urological knowledge and necessary urological skills. The results of the survey were exported and descriptive statistics were performed using IBM SPSS Statistics version 19.0. Results and limitations: A total of 127 answers were obtained, in which ‘Urological infections’, ‘Renal colic’, ‘PSA levels and screening for prostate cancer’, ‘Benign prostatic hyperplasia’, ‘Hematuria’, ‘Scrotal pain’, ‘Prostate cancer diagnosis’, ‘Bladder cancer diagnosis’, ‘Urinary incontinence’, and ‘Erectile dysfunction’ were rated as Very high or High formative requirements (>75%). Regarding urological skills, ‘Abdominal examination’, ‘Interpretation of urinalysis’, ‘Digital rectal examination’, ‘Genital examination’, and ‘Transurethral catheterization’ were assessed as needing Very high or High training in more than 80% of the surveys. The relevance of urological pathology in clinical practice was viewed as Very high or High in more than 80% of the responses. Conclusions: This study has shown helpful results to establish a differentiated prioritization of urological knowledge and skills in Primary Care and Emergencies. Efforts should be aimed at optimizing the teaching in Urology within the Degree of Medicine which consistently ensures patients’ proper care by future GPs.

## 1. Background

One of the main objectives of medical training is to provide students with the tools and knowledge required to diagnose and treat the most common pathologies, regardless of the medical specialty chosen by students. The aging of the population has led to an increase in the prevalence of a number of pathologies, among these, of urological pathology [1]. It is estimated that approximately 40% of US students will opt for generalist specialties and will very likely treat patients with urological problems [2]. Given that General Practitioners (GPs) so frequently act as the future gateway to the Health System, teaching Urology in the Degree in Medicine [1,3] is therefore essential to guarantee that their training equips them with the necessary knowledge and skills to manage these urological patients [4]. In Spain, although 28% of the places offered for specialization after graduation are for GPs, during the 4 years of residency, rotation in urologic units is not mandatory [5].

The standardization of undergraduate medical training across the European Higher Education Area (EHEA) has involved the reduction of the teaching load of the Surgery department in favor of the Medicine department and has, therefore, led to the prioritization of the practical aspect of medical education to ensure adequately qualified future GPs [6,7]. Training time in the Degree of Medicine is limited; thus, some areas might be disadvantaged, as seems to be the case in Urology [8]. Currently, the presence of this subject in the academic undergraduate curriculum is irregular, even non-existent, even when comparing universities in the same country [4]. Data about the Urology teaching quality at universities worldwide are scarce. In Canada, for example, the training exposure to Urology seems to be homogeneously adequate to undergraduate teaching objectives [3], while in the US and the UK, there seems to be a loss in the training quality in Urology. In a study carried out in UK universities, only 37% were found to include Urology in their core curriculum. No formal urology lectures were offered in 31% of them, but rather only practical problem-based cases [9]. Professionals corroborate this lack of training urology competencies during university studies. A survey found that for 65% of students in several US universities, it was possible to finish their medical studies without any exposure to clinical practice in Urology [10]. Likewise, less than a third of the 350 UK urologists participating in another study (27%) admitted having chosen the specialty on the basis of their exposure to the subject during their training in Medicine [8]. Finally, only 9.7% of young physicians in the UK considered the training received in Urology as adequate [11].

The literature has rendered relevant data on the essential role Primary Care in the initial management of urological conditions. GPs play an essential role as the providers of the initial attendance in Health Systems worldwide, also for patients with urological problems. Approximately 3.1% of consultations in general medicine in the US and 5–10% in the UK [12], as well as 20% of the emergencies in the UK, are estimated to be due to urological problems [9]. Hence, the importance of their knowledge in the basic evaluation of these pathologies is determinant for an adequate diagnosis, and consequently, for an optimal treatment of these patients [7,13]. Despite its relevance, concerns have been raised on the lack of evidence on whether GPs are providing worse urological care as a result of the decrease in medical school training time in Urology [2]. By way of illustration, we can mention Nieder et al.’s anonymous survey of 270 GPs, which found that in a large percentage of cases, initial haematuria evaluation would not be correctly addressed [14].

Regardless of the evident differences in the socio-health and economic situation of each country, a common agreement has emerged on the need to develop training programs and curricula which guarantee the acquisition of the necessary competencies for a correct performance of the professional activity which meets the specific needs of each health system [6,7,15]. Using Kern’s [16] six-step approach to curriculum development as a conceptual framework, the aim of our study is to focus on the design of a medical training program which caters for the urological knowledge and skills of future GPs, specifically seeking to respond to the problem identified (Step 1 of Kern’s model) with a needs assessment of urological training needs (Step 2), and on the basis of that, to propose the educational objectives based on the expertise of GPs (Step 3) and performing a curriculum proposal with a theoretical and practical content schedule (Step 4). Although beyond the scope of this paper, the implementation of the program (Step 5) and its evaluation (Step 6) should, in future stages, complement this proposal for curriculum development.

## 2. Methods

The challenges of urological medical education identified allow us to claim that the educational lacks identified need to be addressed, applying curriculum improvement measures. The study presented here departs from Kern’s [16] proposal that communication should be established with stakeholders to assess their needs (Step 2). With that aim, our work seeks to draw on the expertise of GPs to identify the prevailing urological pathologies on the basis of their clinical practice in order to develop improvement proposals for medical teaching and training in Urology.

With this purpose, our study draws on the data of a survey-based study of GPs’ opinions and perceptions to identify the knowledge and skills needed in Urology for the real clinical practice of GPs. An anonymous survey was carried out among GPs of Primary Care and Emergencies in the community of Aragón, Spain (1.318 million inhabitants). The self-administrated 7-min Urological Training Requirements on Primary Health Care Survey were developed and validated in a previous pilot project on 24 Family and Community Medicine 4th-year residents, with 2 years of professional experience in Primary Care; survey reliability was then analyzed using the alpha Cronbach coefficient. The survey sought to elicit the respondents’ views, perceptions and opinions on five issues (Appendix A): (i) the respondents’ profile (items 1–5); (ii) knowledge needs (32 items with 5-answer options); (iii) 5 non-urological knowledge control questions, which ensured the respondents’ coherence in their response; (iv) necessary skills (19 items with 5-answer options); (v) non-urological skills control questions (8 questions with 5-answer options), also used to validate the coherence in responses.

Two versions of the survey were developed, a paper-based one to be sent as an email attachment, and an online one, created using On-Line Survey Generator (Google Drive^®^, Mountain View, CA, USA). The totality of our region Primary Care GPs (1440 GPs) were contacted through their corporate email (access was facilitated by the Regional Health Directorate) together with the 107 Associate Professors in Health Sciences of Primary Care and Emergencies in the Department of Internal Medicine, Psychiatry and Dermatology of the School of Medicine of the University of Zaragoza. Additionally, the GPs in 7 of the Health Service Centers in our community, where the validity of the survey was piloted, were contacted face-to-face. The results of the survey were introduced in the on-line interface, the data were exported, and a descriptive statistics analysis was performed. For data analysis, IBM SPSS Statistics version 19.0 (IBM SPSS Statistics for Windows, Version 19.0. IBM Corp, Armonk, NY, USA) and R language programming version 3.5.2 (The R Foundation for Statistical Computing, Vienna, Austria) were used. Once the items related to the profile of respondents had been excluded, the pre-pilot survey of 24 responses had an alpha Cronbach reliability coefficient 0.93, which showed the internal consistence of the test (Appendix A). Only control items with low correlation appear, thus supporting the validity of the survey design.

## 3. Results

Regarding the entire survey, a total of 127 answers were obtained: 23 from the 107 Associate Professors, 52 collected in Primary Health and Emergency Centers, and 52 from the 1440 Primary Care and Emergency Physicians contacted by email. The demographic and professional characteristics of the respondents are shown in Table 1.

### 3.1. Evaluation of Urological KNOWLEDGE Needs

The five control questions, of non-urological general knowledge, showed a correct coherence in their responses: ‘Headache’, ‘Hypertension’, and ‘Upper respiratory tract infections’, were prioritized with a High or Very high degree of knowledge needed in Primary Care and Emergencies by more than 80% of the participants whereas ‘Huntington’s Chorea’ and ‘Pulmonary transplantation’ were cataloged with a Very low, Low, or Medium degree of knowledge needs by more than 80% of the respondents.

The answers to the objective questions, presented in Figure 1, ‘Urological infections’ (UTI), ‘Renal colic’, ‘PSA levels and screening for prostate cancer’, ‘Benign prostatic hyperplasia’ (BPH), ‘Hematuria’, ‘Scrotal pain’, ‘Prostate cancer diagnosis’, ‘Bladder cancer diagnosis’, ‘Urinary incontinence’, and ‘Erectile dysfunction’ were considered Very high or High formative requirements (>75%) (in decreasing order). Regarding ‘Urological tumors’, the most important aspect considered is diagnosis, above staging and treatment. ‘Infertility’ and ‘Vasectomy’ were regarded as Low knowledge requirements, as well as ‘Congenital disorders’ and ‘Renal transplantation’, pathologies rated with a Very low or Low degree of knowledge requirement.

### 3.2. Evaluation of Urological SKILLS Needs

As for the Control questions, a high level of coherence is observed in the responses obtained. ‘Pulmonary auscultation’ and ‘ECG interpretation’ were cataloged by more than 90% of the respondents as High or Very High in their needs of knowledge for Primary Care and Emergencies practitioners. However, the need to be trained in ‘Laparoscopic cholecystectomy’ or ‘Swan–Ganz catheter placement’ were ranked as Very Low or Low in more than 80% of the responses.

The Objective questions, displayed in Figure 2, highlighted that ‘Abdominal examination’, ‘Interpretation of urinalysis (sediment/culture)’, ‘Digital rectal examination’ (DRE), ‘Genital examination’, and ‘Transurethral catheterization’ were assessed as needing Very high or High training in more than 80% of the surveys (in decreasing order). The rest of the skills, ‘Interpretation of spermiogram’, ‘Evaluation of imaging techniques (urography, CT, resonance, ultrasound)’, ‘Urodynamic tests’, ‘Suprapubic cystostomy catheterization’, ‘Percutaneous nephrostomy placement’, and ‘Cystoscopy’, were considered of Very high or High relevance for training by less than 50% of the respondents (in decreasing order).

### 3.3. Evaluation of the RELEVANCE of Urological Pathology in Their Clinical Practice

As results shown in Figure 3 demonstrate, the relevance of urological pathology in clinical practice was considered Very high or High in more than 80% of the responses. No participant in the survey viewed it as of Very low or Low relevance.

## 4. Discussion

Two major strengths have been highlighted in our study. Firstly, the fact that 75% of the survey respondents have more than 15 years of clinical practice assures well-documented criteria of knowledge and skills necessities in clinical practice. Furthermore, the study was conducted in a EU country, in which features such as health structures, health strategies, and an aged population pyramid entail subsequent necessities in Primary Care. For these reasons, we strongly believe that our proposal could be implemented not only in our context but also in other EU Member States and thus contribute to strengthen quality cooperation in the EHEA.

Following this argument, and based on the aforementioned conceptual framework proposed by Kern [16], the results obtained in the survey allow to us to establish a curricular proposal. This empirical research tool applicable in multiple areas takes shape in this project through this curriculum development, created based on the teaching objectives set out from urological needs in Primary Care. As regards the educational goals and objectives (Step 3) proposed, these are therefore based on the expertise of GPs and on the prevailing urological pathologies managed during their daily clinical practice. The knowledge and skills ranked as Very high or High formative requirements (>75%) were set as the principal objectives. On the basis of these educational goals, our proposal includes the theoretical and practical content schedule summarized in Table 2.

The theoretical contents would be oriented, therefore, towards future practitioners’ general medicine training, grouped into syndromic aspects (haematuria, renal colic...) and in the case of the oncological theoretical contents, 80% of them should be oriented to the diagnosis suspicion and complementary exams, and the remaining 20% to staging and treatment. Regarding the practical contents, the improvement of teaching must be accompanied by a greater practical applicability of that teaching. It not only consists of achieving the integration of the most relevant knowledge and skills to improve a future clinical experience, but also putting it into practice during teaching as a learning method. For this purpose, with the practical workshops summarized in Table 2, we propose this schedule as an example of ‘work-integrated learning’, this powerful pedagogical tool that can facilitate the integration of the theoretical knowledge into clinical practice and thus enable the students to simultaneously learn from theoretical contents and their practical application [17].

Our results ratify claims made in previous publications about undergraduate urological knowledge and skills teaching needs and priorities. Similar results were obtained in Teichman et al.’s online survey administered to directors of General Medicine Residency Programs: the main urological subjects were UTI, sexually transmitted diseases (STD), epididymitis, hematuria, and BPH; the oncological topics were considered less relevant. The diagnostic aspect of the pathologies was viewed as the most relevant necessity, above treatment or staging. Adequate experience in urological examinations, such as DRE or testicular exploration, was rated as of great importance, at the same level of thoracic or abdominal exploration. The respondents also stressed that urological training should be oriented to the training of GPs, and therefore towards appropriate diagnosis and referral to the specialist [18]. Also in Shah et al.’s 41-question survey addressed to urology consultants in the UK, the most relevant knowledge demands were testicular pathology, UTIs and BPH. While the most important practical aspects were testicular exploration, bladder catheterization, DRE, and urinalysis interpretation [12]. Regarding the most relevant contents for basic training in Urology, in a survey addressed to trainers, Kerfoot found that the 6 most relevant needs are urolithiasis, hematuria, UTI, BPH, incontinence, and prostate cancer [19]. These results closely resemble those obtained in our case, highlighting the great importance of the aforementioned pathologies for GPs’ daily practice.

In our context, the program currently being taught is structured as follows: the lessons are structured into expository classes, reinforced by audiovisual tools to complement the training. All this material is available from a repository on an online platform (Moodle is the course management system used by the University of Zaragoza). We have previous experience in the implementation of these and other similar teaching resources [20] which allow us to reach all students in an optimal way, including students with disabilities [21]. We are also experienced in the development of practical workshops (e.g., digital rectal examination workshop) with small groups of students and through the use of simulators, allowing an optimization of the learning of practical contents.

With this, we hope to provide a response to those specialists who have expressed their concern for the scarcity of training in Urology in medical programs and have advocated the development of improvement strategies. In this line, Kerfoot et al. propose a one-week rotation in Urology, which, as their study shows, means a significant difference in knowledge of the main urological pathologies (such as PSA screening or prostate cancer) [22]. In the UK, Miah et al. devised a complementary two-week plan of consolidation training in Urology [9]. Given the constraints of formal training programs, the use of new technologies, such as videoconferences used to teach theoretical contents, interactive programs with multimedia content to complement conventional theoretical training [23,24,25], and simulation-based training programs [26], have been found to significantly improve learning in Urology. In a similar line, the literature has claimed that tools need to be developed to evaluate and improve the teaching of undergraduate Urology, commonly restricted to the evaluation of the training received by final year Degree students [1]. Similarly, constant urological advances, such as the improvement in early diagnosis and in minimally invasive surgery, have also highlighted the need for an evolution in the academic teaching of Urology which ensures that the quality standards remain [7,27].

In sum, this paper claims that it should be desirable to give Urology more relevance in the curricula of medical students. Taking into account the high prevalence of urological diseases in the population, the teaching of Urology deserves a central position in medical training programs [7]. Our project has aimed to understand what knowledge and skills future doctors need to acquire which should inform the prioritization of training goals. Although the time and the opportunity of teaching Urology is variable in the Faculties of Medicine, our study has provided consistent evidence on the training needs of GPs and on how available time should be adapted to meet these necessities.

The results of our survey, although promising, should be interpreted with caution, since the response rate to the survey has been proportionally low: although the totality of our region Primary Care GPs (1440 GPs) were contacted, we received answers from only 127 doctors, of whom 52 were GPs, representing 3.6%. Moreover, surprisingly, only 21.5% of the Associate Professors who should be expected to have a special interest in teaching responded the survey. The dissemination of the online survey has to be mentioned as another limitation: the face-to-face administration of the survey showed itself to be the most effective yet expensive and time-consuming method. Despite limitations, our data have proven particularly helpful for our proposal of the implementation of a Urology curriculum informed by survey data. Similarly, survey-informed decisions can be applied to transfer GPs perceived needs in other specialties of to guarantee that the medical Degree pursues the adequate preparation of future physicians for the real needs they have to face in their professional practice.

Following this concept, and referring again to Kern’s conceptual framework (16), in this project, the last steps, implementation (Step 5) and feedback (Step 6), have not been developed yet. The ultimate goal of this proposal is its inclusion within the study plan in the Degree in Medicine. It is essential to recognize the urological training needs; however, this would be an incomplete effort if no real change in the teaching of Urology actually existed. Therefore, the next modification of the study plan should be built based on proposals like the one we have put forward. In our view, evaluation and feedback should be carried in a longitudinal study in a period of 5 years after the implementation of the new study plan, which would allow the data on the evolution of these students to become solid indicators of its effectiveness.

## 5. Conclusions

This study has yielded helpful results to establish a differentiated prioritization of urological knowledge and skills in Primary Care and Emergencies. Drawing on those, it has sought to offer an interesting reflection on the current situation and to propose an adjustment of the teaching content of the Urology area. It has been noted that urological diseases are of great interest for the Primary Care physicians and therefore efforts should be aimed at optimizing the teaching of the main theoretical content and practical skills in Urology within the Degree in Medicine, ensuring proper care of patients by these GPs in the future.

## Figures and Tables

**Figure 1 ijerph-18-11687-f001:**
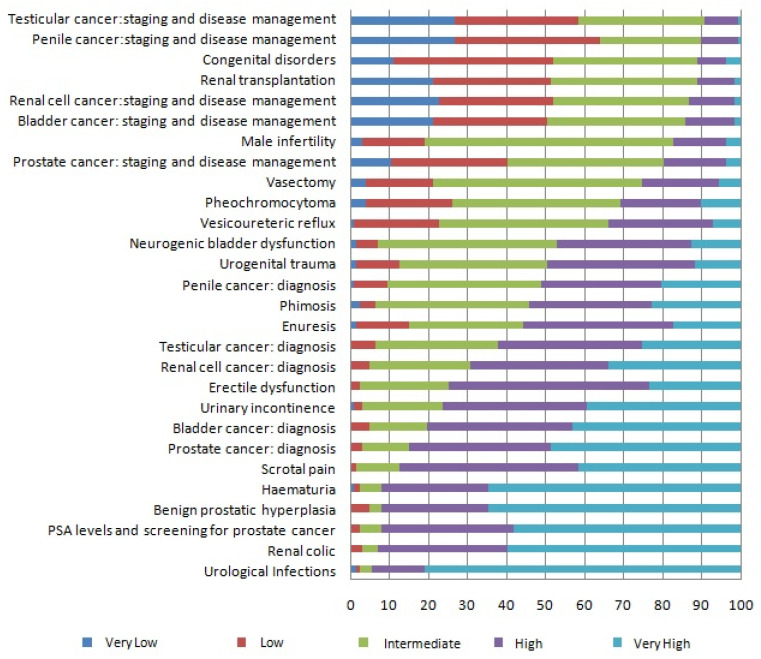
Evaluation of urological knowledge: objective questions.

**Figure 2 ijerph-18-11687-f002:**
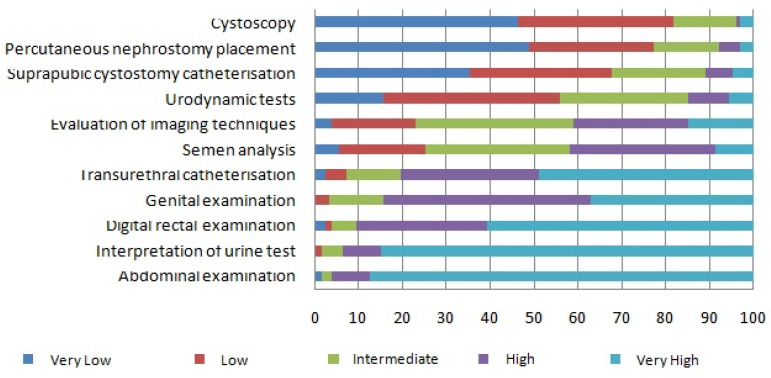
Evaluation of urological competences: objective questions.

**Figure 3 ijerph-18-11687-f003:**
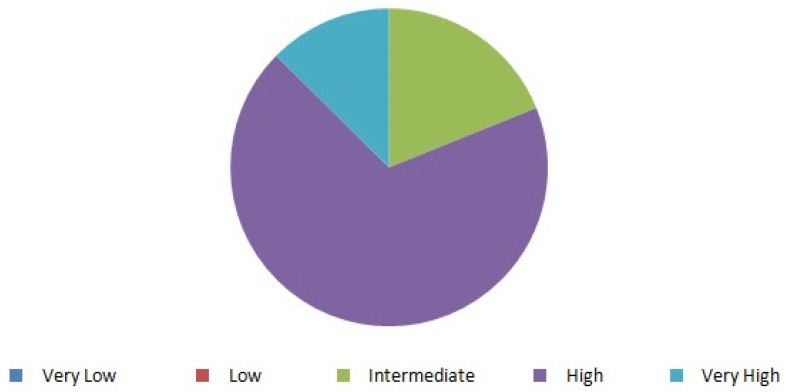
Evaluation of the relevance of urological pathology.

**Table 1 ijerph-18-11687-t001:** Demographic and professional characteristics of respondents.

Years of Professional Practice	
Mean (CI 95%)	22.6 years (20.9–24.2 years)
Median (P25-P75)	23 years (15.25–31 years)
Min-Max	3–38 years
**Sex**	
Men/Women	46.5%/53.5%
**Workspace**	
Rural/Urban	28.3%/71.7%
Primary Care/Emergency	85%/15%
**Associate Professor of Health Sciences**	
Yes/No	23.6%/76.4%

**Table 2 ijerph-18-11687-t002:** Theoretical and practical content schedule.

Theoretical Lessons	Urological infections Urinary incontinence Erectile dysfunction Renal colic Haematuria Scrotal pain Benign prostatic hyperplasia PSA levels & screening for prostate cancer & Prostate cancer Bladder cancer Testicular cancer & Penile cancer Miscellanea: urogenital trauma, congenital malformations, renal transplantation
Practical Workshops	Abdominal examination, Digital rectal examination Genital examination, & Transurethral catheterization Interpretation of urinalysis (sediment/culture)’, and semen analysis

## Data Availability

The datasets used and analyzed during the current study are available from the corresponding author on reasonable request. All data and materials generated or analyzed during this study are included in this published article and its Appendix A.

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
