# Peer review of "Teaching Urology to Undergraduates: A Prospective Survey of What General Practitioners Need to Know"

_ijerph, 2021, doi:10.3390/ijerph182111687_

Round 1

Reviewer 1 Report

To the Editor
I have evaluated the manuscript sent for publication: Teaching urology to undergraduates: A prospective survey of what General Practitioners need to know.
The authors used a survey, previously validated in a pilot group, distributed along with the GP’s of Primary Care and Emergencies in the community of Aragon, in Spain. One hundred answers were obtained (23 from 107 Associated professors, 52 from professionals from Primary Health and Emergency Centers, and 52 from Primary Care and Emergency Physicians. Non-urological knowledge questions were included in the survey questions, as means to evaluate coherence in the answers. Results confirmed the need to optimize reaching Urology in pregraduate medical education.
About the manuscript:
Do not repeat information in tables and figures (select one).
The last paragraph on results would be better placed in the discussion section.
Avoid repetitions
Finally, the information obtained through the survey will be useful in future curriculum modification in medical education, and it is worth publishing provided minor modifications.

Author Response

We want to thank for the insightful comments and suggestions concerning our manuscript. All comments were appreciated and quite helpful in revising and improving our manuscript; the comments also provided important guidance for our research. We carefully reviewed all comments and made revisions that we hope will earn your approval.

I have evaluated the manuscript sent for publication: Teaching urology to undergraduates: A prospective survey of what General Practitioners need to know.
The authors used a survey, previously validated in a pilot group, distributed along with the GP’s of Primary Care and Emergencies in the community of Aragon, in Spain. One hundred answers were obtained (23 from 107 Associated professors, 52 from professionals from Primary Health and Emergency Centers, and 52 from Primary Care and Emergency Physicians. Non-urological knowledge questions were included in the survey questions, as means to evaluate coherence in the answers. Results confirmed the need to optimize reaching Urology in pregraduate medical education.

About the manuscript:
Do not repeat information in tables and figures (select one).

Thank you for your suggestion, we have removed the Tables 2, 3 and 4 in the manuscript to prevent unnecessary repetitions.

The last paragraph on results would be better placed in the discussion section.
Avoid repetitions

We agree with this comment, we have modified the location of the last paragraph of results to the discussion section and we have reviewed the text for the grammar and the repetitions 

Finally, the information obtained through the survey will be useful in future curriculum modification in medical education, and it is worth publishing provided minor modifications.

Thank you for your comment

Reviewer 2 Report

Dear Authors,

your paper highlight an important topic in Urology teaching in Medical Schools and Universities, as urological diseases are many and affect a large part of population (from UTIs to cancer and stone disease). There is just one bias in your paper:

From 1,440 Primary Care GPs, you received answers by only 127 doctors, of whom only 52 were GPs. Thus, your paper results are not an appropriate representation of GP knowledge with only the 3.6% of response rate by them.

Results might be presented anyway, but you must take in account this bias, as a limitation and thus interpreting results. Definitive conclusions cannot be drawn from your study

Author Response

We want to thank for the insightful comments and suggestions concerning our manuscript. All comments were appreciated and quite helpful in revising and improving our manuscript; the comments also provided important guidance for our research. We carefully reviewed all comments and made revisions that we hope will earn your approval.

Dear Authors, your paper highlight an important topic in Urology teaching in Medical Schools and Universities, as urological diseases are many and affect a large part of population (from UTIs to cancer and stone disease). There is just one bias in your paper:

From 1,440 Primary Care GPs, you received answers by only 127 doctors, of whom only 52 were GPs. Thus, your paper results are not an appropriate representation of GP knowledge with only the 3.6% of response rate by them.

We totally agree with this comment, unfortunately our study may be biased due to the low response. Therefore, we have taken the bias into account, and it is specified as a limitation in the text, in the discussion section.

Results might be presented anyway, but you must take in account this bias, as a limitation and thus interpreting results. Definitive conclusions cannot be drawn from your study

Thank you for your suggestion.

Round 2

Reviewer 2 Report

Dear authors,

thank you for taking into account my comments regarding the answer rate bias. Nice paper and idea.